# Influence of Graphene Oxide on the Mechanical Properties, Fracture Toughness, and Microhardness of Recycled Concrete

**DOI:** 10.3390/nano9030325

**Published:** 2019-03-01

**Authors:** Jianlin Luo, Shuaichao Chen, Qiuyi Li, Chao Liu, Song Gao, Jigang Zhang, Junbing Guo

**Affiliations:** 1School of Civil Engineering, Qingdao University of Technology, Qingdao 266033, China; 17853245963@163.com (S.C.); alexlc@163.com (C.L.); gaosong727@126.com (S.G.); zhangjigang_hit@163.com (J.Z.); guojunbing1994@163.com (J.G.); 2Collaborative Innovation Center of Engineering Construction and Safety in Shandong Blue Economic Zone, Qingdao University of Technology, Qingdao 266033, China; 3Center for Infrastructure Engineering, School of Computing, Engineering and Mathematics, Western Sydney University, Sydney, NSW 2751, Australia; 4School of Architecture Engineering, Qingdao Agricultural University, Qingdao 266109, China

**Keywords:** graphene oxide, recycled aggregate concrete, mechanical property, fracture toughness, microhardness, microstructure

## Abstract

There is a constant drive to improve the properties of recycled concrete owing to its inferior strength and fracture toughness compared to normal concrete and recent progress in graphene oxide (GO) nanomaterials impelling nanosized reinforcements to recycled concrete. Here, GO-modified natural sand (NS)- or recycled sand (RS)-based mortars (GONMs or GORMs) with six GO fractions (*w*_GO_s) were fabricated to explore their 28 d mechanical strengths (*f*^28^_t_, *f*^28^_c_), fracture toughness (*K*_IC_, *δ*_c_), and microhardness (*H*_v_), as well as their crystal phases (using X-ray powder diffraction) and microstructures (using scanning electronic microscopy). Results reveal, greater enhancements in mechanical strengths (4.50% and 10.61% in *f*^28^_t_, 4.76% and 13.87% in *f*^28^_c_), fracture toughness (16.49% and 38.17% in *K*_IC_, 160.14% and 286.59% in *δ*_c_), and microhardness (21.02% and 52.70% in *H*_v_) of GORM with just 0.025 wt‰ and 0.05 wt‰ GO, respectively, with respect to the control are achieved when comparing with those of GONM with the same *w*_GO_. More zigzag surfaces, more irregular weak interface slips, and the relatively lower strengths of RS bring the superiority of the template and reshaping effects of GO into full play in GORM rather than in GONM. These outcomes benefit a wide range of applications of recycled concrete products.

## 1. Introduction

Construction waste shows a fast-increasing trend with urbanization and industrialization, and the annual output of construction waste has climbed up to 3.5 billion tons, causing a non-negligible threat to the environment [1,2,3]. Recycling research and technology on construction waste not only conserves many natural aggregate resources but also reduces construction waste pollution, which effectively conforms to the requirements for the sustainable development of the construction industry [4]. Extensive and intensive research on the performance of recycled concrete by incorporating this waste after crushing as a recycled aggregate and/or admixture has been explored all around the world, and its workability, mechanical properties, and durability are found to be generally inferior to those of normal concrete [3,4,5,6,7]. Indeed, owing to recycled concrete being a type of heterogeneous material with multi-phases and multi-interfaces, its microstructure is more complicated than that of normal concrete, and the existence of multiple but weak interfaces is the main reason for the low compressive strength and long-term property of recycled concrete [5,8,9,10]. Therefore, it is necessary to further refine the surface features of recycled aggregate and the microstructures of recycled concrete for the corresponding recycled concrete products with enhanced macro-performances to achieve wider applications. Zhao et al. [11], Sallenhan et al. [12], and Zhu et al. [13] pre-coated recycled aggregate and attempted to improve the drying shrinkage and durability of the corresponding recycled concrete. Several researchers have also doped some fibers or nanoparticles, such as PVA fiber [14], basalt fiber [15], nano-silica [16], nano-SiO_2_, and/or nano-TiO_2_ [17,18] to effectively improve the mechanical properties, pore structure, and chloride diffusivity of recycled concrete. Xiao et al. [18] documented that nano-TiO_2_ was a little better than nano-SiO_2_ for refining recycled concrete, and the corresponding recycled concrete doping with 2 wt% nano-TiO_2_ had the best resistance to chloride diffusion.

As a graphene derivative, a graphene oxide (GO) sheet comprises a hexagonal carbon network bearing hydroxyl, epoxide, carboxyl, and carbonyl functional groups [19]. These oxygen-containing groups contribute to making GO sheets hydrophilic and highly dispersible in water, in contrast to carbon nanotubes [20,21,22]. The various functional groups of a GO sheet and its large surface area make it the perfect nano-reinforcement to aqueous systems due to the improvement in solubility, ease of production, and positive interaction with an aqueous cement matrix [23,24,25,26]. Hou et al. [27] studied the intrinsic interactions between a GO nanosheet and cement hydration products through the reactive molecular dynamics simulation method and found that GO dosage endowed the cementitious matrix with a high cohesive force and enhanced plasticity owing to the H-bonds and covalent–ionic bonds. Quite a few related researchers reported that GO sheet dosage could significantly improve the mechanical properties, fracture toughness, as well as the microstructures of hardened cement concrete [28,29,30,31,32,33,34,35,36,37,38,39,40]. The transport properties, corrosion resistance, freeze-thawing durability, electrical resistivity, multifunctionality/intelligence, and electromagnetic shielding functional capacities of GO-reinforced concrete were also investigated [41,42,43,44,45]. As Mohammed et al. argued, GO could effectively improve the transport property, reshape the microstructure, and enhance the frost resistance of GO-reinforced concrete [41,42]. As Chen et al. found, with cement with 0.4 wt% GO––carbon fiber (CF) and a shield thickness of 5 mm, a shielding effectiveness of 34 dB was attained at the X-band region, a 31% increase over that of CF/cement (26 dB) in the same fraction [43]. In addition, Lu et al. investigated the influence of GO on the workability, hydration degree, mechanical behavior, and microstructures of magnesium potassium phosphate cement (MKPC) paste and concluded that the addition of GO shortened the final setting time and decreased the workability of the MKPC paste, but the mechanical strength of the MKPC paste with the proper addition of GO was improved [44].

However, as Ghazizadeh et al. documented, the direct incorporation of GO nano-reinforcement into a cementitious matrix hardly guaranteed a good dispersion of GO in an alkaline cementitious matrix, and even some agglomeration occurred due to the temporarily retardant interaction of GO with the surface of cement hydrating grains [24]. Li et al. also argued that GO agglomerates, after direct incorporation of GO into the cement matrix, were more electrically insulative than cement paste, and the resultant composite had no piezoresisitive effect [46]. A few scholars endeavored to achieve sufficient dispersion and stabilizing of GO in cement matrix employing ultrasonication [47], polycarboxylate superplasticizer [37,48], and/or silica fume aiding [48]. Here, we use the dispersion techniques of ultrasonication and polycarboxylate superplasticizer pre-stabilization to achieve sufficient dispersion of GO.

To the best of the authors’ knowledge, there are no reports addressing the effect of GO on the fracture toughness and microhardness of recycled concrete. In this study, diluted GO dispersion was firstly treated with the dispersion techniques of ultrasonication and polycarboxylate superplasticizer pre-stabilization; diluted GO dispersions with varied contents of GO were then cast mixed into cement mortars along with natural (NS) or recycled sand (RS) to fabricate six groups of GO-modified NS- or RS-based mortars (GONMs or GORMs). The corresponding mechanical strength, fracture toughness, and microhardness of the 28 d-cured GONMs or GORMs were comprehensively investigated. The reinforcing mechanisms of GO on the physical properties of the RS- or NS-based mortars were also investigated, as well as their crystal phases and microstructures.

## 2. Materials and Experimental Details

### 2.1. Raw Materials

The binder used was Portland cement (P.I. 52.5 type) conforming to standard GB175-2007 (Beijing, China) and obtained from Shanshui Cement Group (Qingdao, China); the main mineral ingredients and physical index is shown in Table 1. The natural sand (NS), river sand obtained from the Qingdao Dagu River, was a class II graduation medium sand with a 2.7 fineness modulus; the sieve results and physical properties are shown in Table 2. Recycled sand (RS) was recycled from the crushed residues of C45 grade waste concrete specimens prior to one-year testing and acquired through a simple crushing and particle reshaping process with a jaw crusher and particle reshaping machine in sequence [1]. The apparent morphology and preparation flow chart of RS is shown in Figure 1 and the physical properties are revealed in Table 3. These effectively meet the requirements of a class II recycled fine aggregate according to standard GB/T 25176-2010 (Beijing, China). The superplasticizer, a polycarboxylate-based high-range water-reducer also used as the dispersant for graphene oxide (GO) in water was obtained from Jiangsu Sobute New Material Co. Ltd. (Nanjing, China). Pristine GO dispersion was acquired from Nanjing Xianfeng Nanomaterials Technology Co. Ltd. (Nanjing, China). Table 4 presents the physico-chemical property index of the superplasticizer and pristine GO dispersion. Water, tap water available, and no coarse aggregate was applicable. Six mix ratios numbered as N1G–N3G and R1G–R3G of GO-modified cement mortars with natural or recycled sand (GONMs or GORMs) are detailed in Table 5, and the superplasticizer dosage (*w*_SP_), water–cement ratio (W/C), and sand–cement ratio (S/C) were respectively set as 1.5%, 0.4, and 2.0 for all mix groups.

### 2.2. Preparation of GONM or GORM Specimens

A total of 30 mL of pristine GO dispersion (1 mg/mL) was first mixed into 80 mL of water and superplasticizer with *w*_SP_ of 1% and ultrasonically dispersed for 1 h at 40 kHz frequency and 60 W power (KH5200 type bath sonicator, Kunshan, China) to prepare a diluted GO dispersion for reservation. The cement mortar slurries of each mix group with NS or RS were fabricated via a mortar mixer with a blade (JJ-5 type, Wuxi, China) after excluding the superplasticizer and water consumption in the resultant diluted GO dispersion according to criterion GB/T17671-1999 (Beijing, China). Then, the resultant diluted GO dispersion with different dosages was mixed into a NS- or RS-based mortar slurry for another 120 s at high speed to prepare GONMs or GORMs in accordance with the mix groups shown in Table 5. 

The flowability of each group slurry was immediately measured with a flow table tester (NLD-3 type, Wuxi, China) after jumping 25 times in accordance with criterion GB/T2419-2005 (Beijing, China) [49]. The photos of the flowability testing of GONM slurry without GO and with *w*_GO_ of 0.05 wt‰ are shown in Figure 2, and the resultant flowability of each group slurry is revealed in Table 5 above. 

Finally, the well-blended slurry for each group was poured into oiled molds to form six prism specimens of 160 mm× 40 mm× 40 mm, association with slurry compact facilitated, and air bubbles inside reduced with a vibrating compact table (ZS-15 type, Wuxi, China) according to criterion GB/T17671-1999 (Beijing, China). All specimens were demolded after 24 h and immersed in water at 20 ± 3 °C for curing to 28 d.

### 2.3. Testing and Characterization Methods

The 28 d-cured flexural strength (*f*^28^_t_) and compressive strength (*f*^28^_c_) of the GONM and GORM specimens were successively measured with a universal material testing machine (DY-208 type, Wuxi, China) at 0.05 MPa/s, 0.5 MPa/s cross-head loading speeds according to GB/T17671-1999 specification. Three and six corresponding half-split duplicates were tested for each group.

First, a round bottom half-depth slot with a 1.0 mm opening at the mid-span of the pristine GONM or GORM specimen was cut on a slot with a keyway-milling machine, and two blade adapters were attached next to both sides of the slot, achieving a single edge notch bend (SENB) layout. A digital electrohydraulic servo testing machine (WD-9403C type, Shimadzu, Japan) and an electronic extensometer with a gauge length of 5 mm (YYU-5/50 type, Beijing, China) recorded the loads (*P*s) and the crack mouth opening displacements (*CMOD*s) of the notch specimen at 0.05 mm/min loading speed and 0.1 N load cell [29,39,50,51]. The SENB configuration of fracture toughness testing and the real-time data acquisition system for three duplicate specimens in each group are presented in Figure 3. The corresponding fracture toughness (stress-intensity factor *K*_IC_) of the GONM or GORM specimen can be calculated by the following Equations [50]:(1)$$K_{IC}=\frac{P_{\max}S}{BW^{3/2}}f(\frac{a}{W})$$
(2)$$f(\frac{a}{W})=2.9{\left(\frac{a}{W}\right)}^{1/2}-4.6{\left(\frac{a}{W}\right)}^{3/2}+21.8{\left(\frac{a}{W}\right)}^{5/2}-37.6{\left(\frac{a}{W}\right)}^{7/2}+38.7{\left(\frac{a}{W}\right)}^{9/2}$$
(3)$$B\geq 2.5(\frac{K_{IC}}{f_{t}^{28}}{)}^{2}$$
where *S* is the span (m), *W* is the height (m), *B* is the width of the specimen (m), *a* is the mean depth of the slot (m), *P*_max_ is the peak load (N), and *f*^28^_t_ is the 28 d flexural strength (Pa) of the pristine specimen. Note that only when meeting Equation (3) can the *K*_IC_ value be deemed to be effective.

After surface polishing to GONM or GORM half-split block following flexural strength testing with 240, 320, 600, and 1200 mesh sand papers, polishing cloth, and polishing agent in sequence, the microhardness values (*H*_v_s) of GONM and GORM within a 4 × 5 lattice box were measured by a Vickers microhardness tester (HX-1000T type, Laizhou, China) under 0.05 kgf load with a 10 s compression duration in accordance with standard ASTM E384-08 (Philadelphia, PA, USA) [52]. It is worth noting that in order to reflect the indentation gradient of *H*_v_ in the interface region, the height difference between each adjacent two points within the 4 × 5 lattice box was set to 10 μm, as demonstrated in Figure 4 [4]. The corresponding *H*_v_ of the GONM and GORM specimen could be derived by Equations (4) and (5):(4)F=d22sinθ
(5)Hv=P/F=P×2sinθd2=1.8544P/d2
where *F*, *d*, *θ*, and *P* is the notch area (μm^2^), the diagonal length (μm) of the rectangular pyramid indentation, the angle between the notch head and the specimen surface (°), and the applied load (N) of the specimen, respectively.

X-ray powder diffraction (XRD, Bruker D8 Advance type, Leipzig, Germany) and scanning electronic microscopy (SEM, FEI Helios NanoLab 600 type, Hillsboro, OR, USA) were further used to analyze the crystalline structures and microstructures of the tiny GONM or GORM samples after mechanical crushing and oven-drying at 60 ± 0.5 °C for 48 h.

## 3. Results and Discussion

### 3.1. Workability and Mechanical Properties

As shown in Table 5, the viscosity of GONM slurry with high GO loading dramatically increases with respect to the control, and the flowabilities of GONMs or GORMs show significant reductions, but those of all groups are more than 120 mm, which effectively meets the workability requirements of criterion GB/T2419-2005 (Beijing, China).

Table 6 demonstrate the *f*^28^_t_s, *f*^28^_c_s, and the corresponding enhancements of the three groups of GONM and GORM specimens with respect to those baselines. As revealed in Table 6, both the *f*^28^_t_ and *f*^28^_c_ of GONM (5.81 MPa, 50.6 MPa and 6.15 MPa, 55.0 MPa, respectively) are steadily higher than those of GORM (6.01 MPa, 55.3 MPa and 6.35 MPa, 59.5 MPa) with the same GO fraction (*w*_GO_ at 0.025 wt‰, 0.05 wt‰). Moreover, the *f*^28^_t_ and *f*^28^_c_ of GONM or GORM with GO are all relatively higher than those without GO, and the enhancements in *f*^28^_t_ and *f*^28^_c_ of GORM are a little higher than those of GONM with the same *w*_GO_. 

### 3.2. Fracture Toughness Properties

The *P*-*CMOD* curves associated with the secant OP_5_ of GONM and GORM specimens with varied *w*_GO_s are shown in Figure 5. It’s noting, the slopes of secant OP_5_ curves are 95% of those of the initial tangents of the *P*-*CMOD* curves. The corresponding fracture toughness properties (*K*_IC_, *δ*_c_), after being verified by Equation (3), are listed in detail in Table 7.

The *K*_IC_s of GONM are all higher than those of GORM with the same *w*_GO_, the *K*_IC_s and *δ*_c_s of GONM with GO are higher than those without GO, and the maximum enhancements are 16.27%, 41.90%, 22.21%, and 59.11% when the corresponding *w*_GO_ is only 0.025 wt‰ and 0.05 wt‰, respectively. Whereas the *K*_IC_s and *δ*_c_s of GORM with *w*_GO_ of 0.025 wt‰ and 0.05 wt‰ reach 0.6019 MPa.m^1/2^, 0.1397 and 0.7139 MPa.m^1/2^, 0.2076, respectively, which is much higher than those without GO (0.5167 MPa.m^1/2^, 0.0537), the corresponding increases are up to 16.49%, 160.14% (*w*_GO_ at 0.025 wt‰) and 38.17%, 286.59% (*w*_GO_ at 0.05 wt‰) with respect to those of the baseline. It’s clear that the enhancement amplitudes in *K*_IC_ and *δ*_c_ of GORM are all higher than those of GONM with the same *w*_GO_.

### 3.3. Microhardness

As revealed in Figure 6, there are discrete degrees in the *H*_v_s of GORMs and GONMs, and relatively higher deviations in the *H*_v_s of GORMs than GONMs with the same *w*_GO_. Therefore, a notch box diagram is employed to remove outliers in the point data, using the upper quartile and the next quartile to determine the *H*_v_s within the standard area of GONMs and GORMs. As plotted in Figure 6, the *H*_v_ means of GONMs with 0 wt‰, 0.025 wt‰, and 0.05 wt‰ GO are 69.3 kgf, 85.8 kgf, and 104.7 kgf, respectively, and are steadily higher than those of GORMs with the same *w*_GO_ (68.5 kgf, 82.9 kgf, and 104.6 kgf, respectively). It is worth noting that the *H*_v_ mean of GORM with 0.05 wt‰ *w*_GO_ is almost equal to that of GONM, implying greater enhancement in the *H*_v_ mean for GORM.

### 3.4. XRD and SEM Analysis

Figure 7 presents the crystal phases of GONM and GORM with varied *w*_GO_s. 

As revealed in Figure 7, the crystal phases of GONM and GORM with GO mainly comprise of ettringite (AFt), calcium hydroxide (CH), calcium polysilicate, and calcium silicate hydrate (C-S-H), which are similar to those without GO [5]. Only the main peaks of calcium polysilicate of GONM with 0.025 wt‰ and 0.05 wt‰ GO (33.45, 36.07, respectively) are generally higher than those without GO (26.10). The peak of AFt is dramatically reduced after GO incorporation, and especially that of GONM, as shown in Figure 7a,b. Furthermore, the main peak of GONM is higher than that of GORM with the same *w*_GO_, and higher *w*_GO_ incorporation brings forth a higher peak gap between GONM and GORM, which implies a relatively inferior crystalline quality of cement hydrations in GORM than in GONM. Obviously, GO dosage can render cement hydration product to be deposited and grown on GO sheets, but no new crystals emerge, and GO incorporation can only promote the pattern and size regulations of cement hydrations [31].

Unfortunately, GO is almost identical to the morphology of cement hydration in both color and form, and it is almost impossible to point it out and distinguish it from the microstructure. Here, we compare the morphology of cement hydrations before and after incorporating GO to achieve indirect proof of the effect of GO on microstructure improvements, as is also reported in the literature [20,28,30,32,35,38,48]. As shown in Figure 8a,b, many stab-like CH crystals and some needle-like AFt random deposits in hydrated C-S-H gels, and a number of capsular micropores with irregular sizes, contribute to the loose texture of hydration products, which results in the relatively low strength, fracture toughness, and microhardness of GONM or GORM without GO. Whereas the C-S-H gels become prevailing, appearing much bigger and denser [30], and very few CH and AFt crystals can be found in the GONM or GORM with 0.05 wt‰ GO, the improved macro-performances are accordingly achieved owing to the template and nucleating effect of GO to cement hydration, as presented in Figure 8c,d.

### 3.5. Improvement Mechanisms for the Mechanical Strength, Fracture Toughness, and Microhardness of GONM or GORM

Waste concrete used to produce RS was collected from C45 grade crushed concrete specimens in our lab, and the inside aggregate hardly suffered from any weathering and environmental corrosion after one-year exposure. It retained its integrity, but there inevitably exist some micro-cracks in RS and some old hydration product attached to RS even after rough jaw crushing and reshaping treatment, which harm the interface bonding between RS and new cement hydration products (presented in Figure 1). This results in moderate deteriorations in the mechanical strengths (*f*^28^_t_, *f*^28^_c_), fracture toughness (*K*_IC_, *δ*_c_), and microhardness (*H*_v_) of GORM with respect to those of GONM.

As a graphene derivative, GO sheets have hydroxyl, epoxide, carboxyl, and carbonyl oxygen-containing functional groups, which are hydrophilic and highly reactive nano-reinforcements in aqueous system. The active ingredients in cement grain will be absorbed and assembled on GO sheets, the hydration reaction will firstly occur on the active groups of GO sheets, inducing cement hydration on the active groups of GO sheets to grow into regular micro-crystal products. Size-regular and overlapping hydration products throughout the mortar can be achieved owing to the obvious template and reshaping effects of GO after sufficient dispersion by ultrasonication and polycarboxylate-based superplasticizer stabilization [20,23,30], which are also verified by SEM microstructures (Figure 8). These effects result in significant improvements in the mechanical strengths (9.48% in *f*^28^_t_, 12.90% in *f*^28^_c_), fracture toughness (22.21% in *K*_IC_, 59.11% in *δ*_c_), and microhardness (51.08% in *H*_v_) of GONM only at an extremely low fraction of GO (0.05 wt‰).

Furthermore, greater enhancements in mechanical strengths (4.50% and 10.61%, respectively, in *f*^28^_t_, 4.76% and 13.87%, respectively, in *f*^28^_c_), fracture toughness (16.49% and 38.17%, respectively, in *K*_IC_, 160.14% and 286.59%, respectively, in *δ*_c_), and microhardness (21.02% and 52.70%, respectively, in *H*_v_) of GORM with 0.025 wt‰ and 0.05 wt‰ GO with respect to the baseline are achieved when comparing with those of GONM with the same *w*_GO_. More surrounding zigzag surfaces, irregular interfacial slips between recycled aggregate and surface-coating old mortar owing to inevitable inner micro-cracks, and the relatively lower strengths of RS bring the superiority of the template and the reshaping effects of GO into full play in GORM rather than in GONM (as shown in Figure 8).

## 4. Conclusions

In this study, six groups of GO-modified cement mortars associated with natural or recycled sand (GONMs or GORMs) are prepared, and their mechanical strength, fracture toughness, and microhardness in tandem with microstructures are accordingly characterized.

1) The *f*^28^_t_ and *f*^28^_c_, *K*_IC_ and *δ*_c_, and *H*_v_ of GORM with 0.05 wt‰ *w*_GO_ are 6.15 MPa and 55.0 MPa; 0.7139 MPa.m^1/2^ and 0.2076; and 104.6 kgf, respectively, whereas those of GONM with same *w*_GO_ are just 6.35 MPa and 59.5 MPa; 0.7671 MPa.m^1/2^ and 0.1276; and 104.7 kgf, respectively. This is attributed to RS with zigzag surfaces, irregular interface slips, and relatively inferior strengths bringing the superiority of the template and the reshaping effects of GO into full play in GORM.

2) RS-based mortars with only 0.025 wt‰ and 0.05 wt‰ GO dosage achieve significant improvements in mechanical strengths (4.50% and 10.61% in *f*^28^_t_, respectively, and 4.76% and 13.87% in *f*^28^_c,_ respectively), fracture toughness (16.49% and 38.17% in *K*_IC_, respectively, 160.14% and 286.59% in *δ*_c,_ respectively), and microhardness (21.02% and 52.70% in *H*_v,_ respectively) with respect to the control without GO.

3) XRD and SEM images demonstrate that GO incorporation can only improve the overlapping pattern and size-regular morphology of hydration products, rather than introduce new crystal phases into cementitious matrix.

These findings effectively show a promising future for the use of small amounts of GO to reinforce recycled concrete, and we will focus on the influence of GO incorporation on the transport properties and durability of recycled concrete in the future.

## Figures and Tables

**Figure 1 nanomaterials-09-00325-f001:**
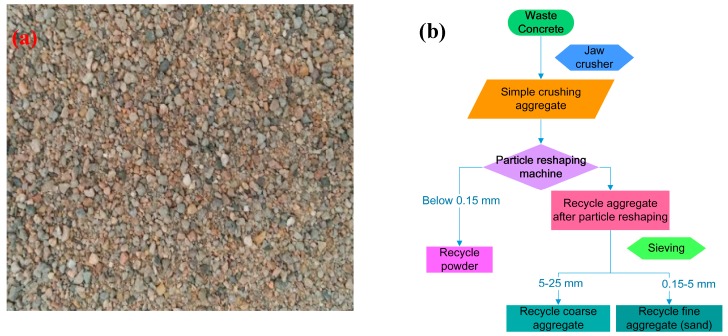
Recycled sand: (**a**) apparent morphology; (**b**) flow chart of recycled sand (RS) preparation with a simple crushing and particle reshaping process.

**Figure 2 nanomaterials-09-00325-f002:**
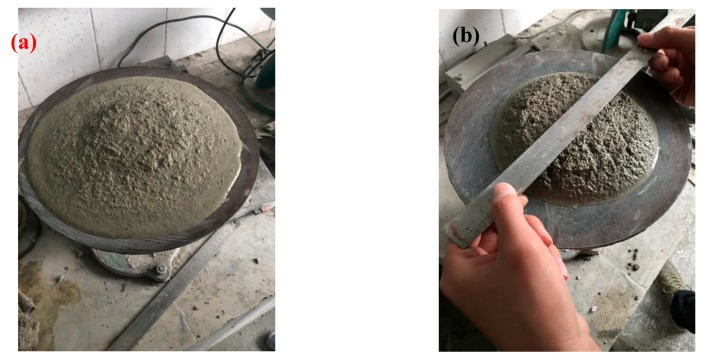
Photos of flowability testing: (**a**) GO-modified natural NS-based mortar (GONM) slurry without GO; (**b**) GONM slurry with *w*_GO_ = 0.05 wt‰.

**Figure 3 nanomaterials-09-00325-f003:**
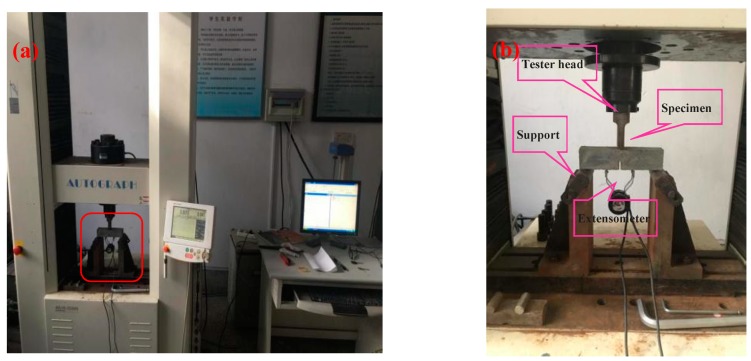
(**a**) The configuration of fracture toughness testing and the Shimadzu data acquisition system; (**b**) the function layout for the local-enlarged red round-angle box of (**a**).

**Figure 4 nanomaterials-09-00325-f004:**
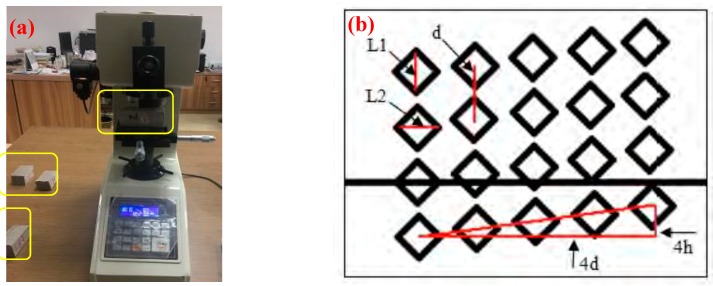
(**a**) The microhardness toughness testing setup (yellow squares: split specimen blocks after surface polishing by 240, 320, 600, and 1200 mesh sand papers, polishing cloth, and polishing agent in sequence; (**b**) schematic claim of microhardness point group.

**Figure 5 nanomaterials-09-00325-f005:**
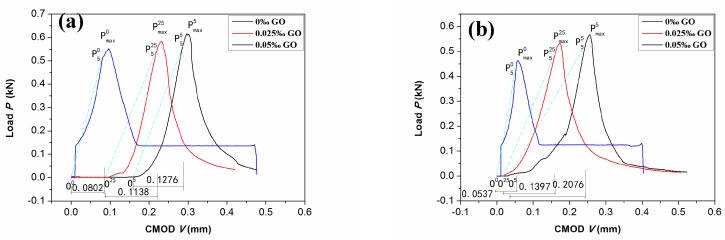
The *P*-*CMOD* curves of (**a**) GONM; (**b**) GORM.

**Figure 6 nanomaterials-09-00325-f006:**
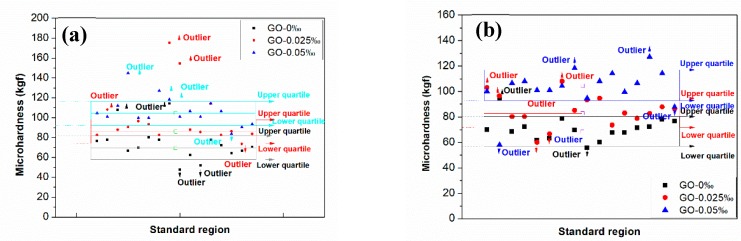
Microhardness distributions in a standard region of (**a**) GONM; (**b**) GORM specimens with varied *w*_GO_.

**Figure 7 nanomaterials-09-00325-f007:**
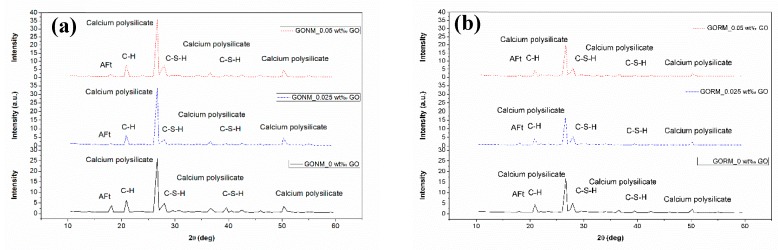
XRD crystalline analysis: (**a**) GONM without GO and with *w*_GO_ = 0.025 wt‰ and *w*_GO_ = 0.05 wt‰; (**b**) GORM without GO and with *w*_GO_ = 0.025 wt‰ and *w*_GO_ = 0.05 wt‰.

**Figure 8 nanomaterials-09-00325-f008:**
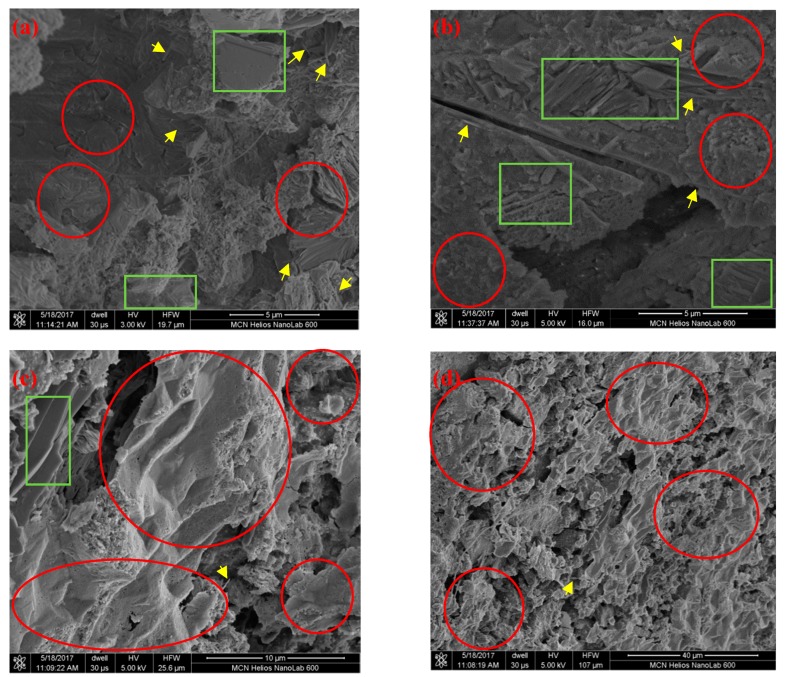
SEM images: (**a**) GONM without GO; (**b**) GORM without GO; (**c**) GONM with *w*_GO_ = 0.05 wt‰; (**d**) GORM with *w*_GO_ = 0.05 wt‰ (green rectangle: CH crystal; yellow arrow: AFt; red circles: C-S-H gel).

**Table 1 nanomaterials-09-00325-t001:** The main mineral ingredients and physical index of cement.

	**Comp.(wt%)**	**SiO_2_**	**Al_2_O_3_**	**CaO**	**MgO**	**SO_3_**	**Fe_2_O_3_**	**K_2_O**	**Na_2_O**	**Loss on Ignition**
**Type**										
P.I. 52.5		20.87	4.87	64.47	2.13	2.52	3.59	0.65	0.11	0.77
	**Phys. Index**	**Density (g/cm^3^)**	**Brunauer-Emmett-Teller Surface (g/m^2^)**	**Soundness**	**Setting Time (min)**		**Flexural Strength (MPa)**		**Compressive Strength (MPa)**	
**Type**					Initial	Final	3 d	28 d	3 d	28 d
P.I. 52.5		3.15	458.0	Qualified	175	255	4.0	7.0	23.9	52.5

**Table 2 nanomaterials-09-00325-t002:** Sieve analysis of natural sand (NS).

Sieve size (mm)	4.75	2.36	1.18	0.6	0.3	0.15	Sieve Bottom
Cumulative screening (%)	4.6	16.8	33	59.8	89.6	99.2	99.8

**Table 3 nanomaterials-09-00325-t003:** Physical property index of class II recycled sand (RS).

Fineness Modulus	Apparent Density (kg/m^3^)	Bulk Density (kg/m^3^)	Water Absorption (%)	Clay Content (%)	Crushing Index (%)
2.70	2503	1469	49–56	41–50	22.3

**Table 4 nanomaterials-09-00325-t004:** Physico-chemical properties of superplasticizer and pristine graphene oxide (GO) dispersion.

**Superplasticizer**	**Solid Content (%)**	**pH Value**	**Water Reduction (%)**	**Air-entraining Content (%)**	**Alkali Content (%)**
Polycarboxylate-type	30	6.5–8	25–35	2–5	≤0.2
**Pristine** **GO Dispersion**	**Concentration**	**Monolayer Content**	**Dispersant**	**Flake Diameter (nm)**	**Oxygen Content (%)**
XF020	1 mg/mL	>95%	water	<500	41–50

**Table 5 nanomaterials-09-00325-t005:** Mix design and testing workability of GO-modified mortar associated with NS or RS (assuming the cement content is 1).

	wt%	Water	Superplasticizer	NS	RS	GO	Flowability (mm)
Mix No.							
N1G		0.40	0.015	2.0	0	0	187
N2G						0.0025	166
N3G						0.005	128
R1G				0	2.0	0	182
R2G						0.0025	159
R3G						0.005	121

**Table 6 nanomaterials-09-00325-t006:** Mechanical strengths of six groups of GONM and GO-modified RS-based mortar (GORM) specimens.

*w*_GO_ (wt‰)	*f*^28^_t_ for NS (MPa)	*f*^28^_t_ for RS (MPa)	Enhancement for NS/RS (%)	*f*^28^_c_ for NS (MPa)	*f*^28^_c_ for RS (MPa)	Enhancement for NS/RS (%)
0	5.80	5.56	-/-	52.7	48.3	-/-
0.025	6.01	5.81	3.62/4.50	55.3	50.6	4.93/4.76
0.05	6.35	6.15	9.48/10.61	59.5	55.0	12.90/13.87

**Table 7 nanomaterials-09-00325-t007:** Fracture toughness properties of six groups of GONM and GORM specimens.

Mix No.	*f*^28^_t_ (MPa)	*P*_max_ (kN)	*a* (mm)	Initial *a/W* ratio	*f*(*a/W*)	*K*_IC_ (MPa.m^1/2^)	Δ*K*_IC_ (%)	Critical *CMOD* (*δ*_c_)	Δδc (%)
N1G	5.80	0.55129	18.355	0.4589	3.6441	0.6277	-	0.0802	-
N2G	6.01	0.58247	19.330	0.4833	4.0097	0.7298	16.27	0.1138	41.90
N3G	6.35	0.61583	19.270	0.4818	3.9864	0.7671	22.21	0.1276	59.11
R1G	5.56	0.46375	18.135	0.4534	3.5656	0.5167	-	0.0537	-
R2G	5.81	0.53283	18.275	0.4569	3.6154	0.6019	16.49	0.1397	160.14
R3G	6.15	0.56725	19.375	0.4844	4.0273	0.7139	38.17	0.2076	286.59

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
