# Peer review of "Influence of Graphene Oxide on the Mechanical Properties, Fracture Toughness, and Microhardness of Recycled Concrete"

_nanomaterials, 2019, doi:10.3390/nano9030325_

Round 1
Reviewer 1 Report
I have evaluated the manuscript entirely
First of all,introduction should be developed. It should be given more detail about the main goal and what is the new in this manuscript regarding to the former ones in the literature.
The authors should givedetail info about their methodology And finally, the conclusion should be improved
Author Response
Dear Reviewer,
We would like to express our sincere appreciations for your valuable comments and suggestions to improve the quality of this paper. The responses are mentioned as below, and the corresponding revisions are highlighted and tracked in the revised edition.
With best wishes,
Jianlin Luo
On behalf of the author team

Reviewer 2 Report
Comments:
In this paper, the authors have utilized graphene oxide to Improve Mechanical Properties, Fracture Toughness, and Microhardness of Recycled Concrete”. The authors are invited to consider the following points in revising their paper. The most important part is English writing, which is not acceptable and the overall manuscript must completely be modified. The paper might be accepted after major revision:
Title:
The title of the paper does not seem appropriate. Especially the initials “using graphene oxide”. Page 1, Line 2-4.
Abstract:
In this section, author just discussed the experimental outcome. A summary that identifies the purpose, problem, methods, trends, results and conclusion of your work is expected in this section”. Page 1, Line 19-28
Materials and Experimental Details: Raw Materials:
Repetition of word “NS” in Table 5. Correct it. Page 4, Line 103.
Materials and Experimental Details:
Preparation of GONM or GORM specimens: “The slurry’s flowability was measured with a jumping table (NLD-3 type, Wuxi,China) resulting of more than 120 mm after jumping 60 s, even though the GONMs or GORMs slurry with GO loading exist some reduction on the flowability, which meets the workability requirement”. First, replace the words jumping table with flow table and jumping with dropped and raised. Also, what’s the criterion for workability requirement here??? Page 4, Line 112-114.
Materials and Experimental Details: Testing and Characterization methods:
“Microstructures of the GONM or GORM tiny sample after mechanical crushing and oven-dry for 48 h, respectively”. What’s the drying temperature and why 48 hours? Page 6, Line 155-156.
6. Results & Discussions: Fracture Toughness properties:
“The KICs of GONM are all higher than those of GORM with the same wGO, the KICs and δcs of GONM with GO are higher than those without GO” “It’s clear that, the enhancement amplitudes in KIC and δc of GORM are all higher than”. These two statements are contradictory. Correct them. Page 7, Line 169-170 & Line 175-176.
7. Results & Discussions: Crystalline and microstructure:
First of all, change the sub-section title “crystalline and microstructure” with a suitable one. Page 8, Line 186
8. Results & Discussions: Crystalline and microstructure:
“Only the peaks of calcium polysilicate of GONM or GORM with GO are generally higher than those without GO, as shown in Figure 6 a) & b).” However, from the figure 6a and 6b, the difference in peaks of calcium polysilicate is not significant for all mixes, or it looks same. Page 8, Line 190-191
9. Results & Discussions: Crystalline and microstructure:
“As shown in Figure 7 a), b), lots of stab-like C-H crystals and some needle-like AFt random deposit in hydrated C-S-H gels, and a number of capsular micropores with irregular sizes contribute to loose texture of hydration products, whereas the C-S-H gels become prevailing, showing much bigger and denser [30], and very few C-H and AFt crystals can be found in the GONM or GORM with 0.05 wt‰ GO, as presented in Figure 7 c), d)”. The justifications provided from SEM images are inadequate. Highlight the portion on SEM images to justify the microstructure improvement claim. Page 8, Line 195-199

Author Response
Dear the Reviewer,
We would like to express our sincere appreciations for your valuable comments and suggestions to improve the quality of this paper. The responses are mentioned as below, and the corresponding revisions are highlighted and tracked in the revised edition along with substantial improvements on English language and styles.
With best wishes,
Jianlin Luo
On behalf of the author team

Round 2
Reviewer 2 Report
The authors have revised the manuscript titled “Influence of Graphene Oxide on Mechanical Properties, Fracture Toughness, and Microhardness of Recycled Concrete”. There are still some points unanswered as the provided justifications are inadequate. The authors are invited to consider the following points in revising their paper. The paper might be accepted after major revision:
1- Materials and Experimental Details: Raw Materials:
“Table 4 presents the physco-chemical property index of superplasticizer and GO pristine dispersion”.
Firstly, physco-chemical is a wrong word. It should be physico-chemical. Secondly, GO-pristine dispersion isn’t correct. It should be pristine GO dispersion. Page 3-4
“Table 5. Mix design of GO modified mortar assocated with NS or RS (assuming the cement content is 1)”. Associated is spelled wrong. Check for spellings in revised manuscript. Page 4.
2- Materials and Experimental Details:
Preparation of GONM or GORM Specimens:
“The flowability of each group slurry was immediately measured with a flow table and jumping with dropped and raised (NLD-3 type, Wuxi, China), all resulting of more than 120 mm after jumping 60 s. even though the GONMs or GORMs slurry with high GO loading exist some reduction on the flowability, which still meets the workability requirement of GB/T2419-2005 criterion (China)”.
Firstly, the flow table test or flow test isn’t described correctly. Revised it and avoid redundant words. Secondly, please provide accurate measurement of workability (calculated flow diameter) if available. Page 5.
3- Materials and Experimental Details: Testing and Characterization Methods:
“Microstructures of the GONM or GORM tiny sample after mechanical crushing and oven-dry at 70±0.5 ℃ for 48 h, respectively”.
The authors mentioned in cover letter (response 5) that the drying temperature for the specimen was set at 60±0.5°C in a vacuum oven, and oven-dry for 48 h, which is the same protocol as Ref. [30] adopted. But these temperature values are contradictory to the manuscript value of 70±0.5°C. Page 7.
4- Results & Discussions: XRD and SEM Analysis:
“Only the main peaks of calcium polysilicate of GONM or GORM with GO are generally higher than those without GO”. However, from the figure 6a and 6b, the difference in peaks of calcium polysilicate is not significant for all mixes. Page 9.
5- Results & Discussions: XRD and SEM Analysis:
“As shown in Figure 7 a), b), lots of stab-like C-H crystals and some needle-like AFt random deposit in hydrated C-S-H gels, and a number of capsular micropores with irregular sizes contribute to loose texture of hydration products, which render relatively low strength, fracture toughness, and microhardness of GONM or GORM without GO. Whereas the C-S-H gels become prevailing, showing much bigger and denser [30], and very few C-H and AFt crystals can be found in the GONM or GORM with 0.05 wt‰ GO, the improved macro-performances are accordingly achieved owing to the template and nucleating effect of GO to cement hydration, as presented in Figure 7 c), d)”.
What is meant by “owing to the template”? Also, the justifications provided from SEM images are inadequate. Highlight the area/portion on SEM images to justify the microstructure improvement claim with GO. Page 9-10.
Author Response
Dear the Reviewer,
We are obligated to your careful reading and comments. The responses are detailed mentioned as below, and the corresponding changes are highlighted and tracked in the 2nd revised edition along with English spelling checking and improvements throughout the paper.
With best regards,
Jianlin Luo
On behalf of the author team

Round 3
Reviewer 2 Report
The authors have modified the paper according to this reviewer's comments- Publication can be recommended.